# Sustainability at an Urban Level: A Case Study of a Neighborhood in Dubai, UAE

**Sundus Shareef [1,*] and Haşim Altan [2,\*]**

1 Faculty of Engineering & IT, British University in Dubai (BUID), Dubai 345015, United Arab Emirates
2 Faculty of Design, Arkin University of Creative Art and Design (ARUCAD), Kyrenia 99300, Cyprus
* Correspondence: sundus.l.shareef@gmail.com (S.S.); hasimaltan@gmail.com (H.A.);
  Tel.: +971-50-638-9133 (S.S.); +90-533-835-9090 (H.A.)

**Abstract:** The United Arab Emirates is witnessing enormous growth and the sustainability attitude has become one of the most important priorities in this development. This paper aims to optimize the environmental sustainability of the Emirate of Dubai communities by adopting an existing community as a case study. The investigation of the case study is looking at sustainability levels that consists of two major factors in neighborhood sustainable design, such as livability and thermal performance. The strategy of enhancing and optimizing the communities' sustainability starts with an approach to the applicable modifications and solutions to the existed community master planning, where the modifications cover the two main urban design variables; (a) building design, and (b) open and landscape areas. The effect of the adopted scenarios is analyzed to find the improvement in environmental and thermal performance. The study has adopted two computer software packages, namely CityCAD and Integrated Environmental Solutions—Virtual Environment (IES-VE), to undertake the assessments. Furthermore, factors of urban sustainability are evaluated using the United States Green Building Council (USGBC)'s Leadership in Energy and Environmental Design (LEED) neighborhood assessment tool. The results have shown that the environmental sustainability levels can be increased after the adoption of certain suggested scenarios, in order to mitigate the likely weakness indicated in the livability aspects, covering land-use diversity, accessibility, transportation system, green and landscape areas, and energy efficiency, and the case study community can be turned toward "Sustainable Community" by implementing recommended actions and modifications.

**Keywords:** sustainability and livability of neighborhoods; sustainable urban environments; sustainable solar shading; building height diversity; United Arab Emirates

## 1. Introduction

Cities are numbers of communities and neighborhoods where people can work live and have entertainment. Day by day cities offer tremendous opportunities for community, employment, education, excitement and interest. For all of these reasons, cities became attractive areas for living and more than half of the world's population are living in cities [1]. On the other hand, cities create problems of congestion, noise, and pollution, but most people do not have the choice, recognizing the trade-offs. How to live and getting the right balance are parts of the solution. Living in towns and low-density cities has some advantages, however people may like living in a compact and dense city as far as there is an equilibrium among the development elements; built area and open spaces, private and public transportation, using natural and artificial resources [2]. City, community or neighborhoods could be considered as a system of depending components [3]. The major variables or components that affect the design of any development are; urban form, transport, landscape, building design, waste management, energy and water supply. The most sustainable design is about equilibrium among these components [4]. In order to make cities or neighborhoods more suitable for people, all aspects of viable city and neighborhoods are required to involve and operate smoothly within design or system equation.

It is obvious that communities and city growth becomes a key issue in the global problems of climate change, global warming, greenhouse gas emissions, and depleting natural resources. Therefore, many studies and publications have explored the relationship between the sustainability levels required to be achieved and the urban planning of any development. Many of these studies concentrated on the main urban design factors, such as urban form, building design, liveability, land use, and transportation system to analyse, evaluate and develop the sustainability level of cities and developments [5,6]. However, some of these researchers studied sustainability on an urban scale from the aspect of resource conservation and pollution reduction. Cities and urban environment pollution are caused by different factors; density and transport within cities, human activities, construction and buildings' effects on nature and landscape areas, atmospheric pollution by CO2 emissions, and noise pollution [7]. Pollution influences human health and well-being and can make cities uneasy places to live. Greenhouse gas (GHG) averages constitute one of the most used air pollution indicators. Greenhouse gas emissions refer to all gases that trap heat in the atmosphere; the main greenhouse gases in the atmosphere are; carbon dioxide (CO2), methane (CH4), and nitrous oxide (N2O). These gases are the main reason for global warming, depleting the ozone layer, and climate change [8]. GHG are emitted through various fossil fuel burning processes. Fossil fuels (coal, natural gas, and oil) are burnt for heating, solid waste burning, trees, and wood products; the decay of organic waste in municipal solid waste landfills, chemical reactions, and manufacturing operations are some resources of GHG [9].

Furthermore, the building and construction industry, transportation, agriculture, and industry are the major recourses of these gases. Global warming, urban heat island (UHI), and the increase in global air temperature are a result of GHG emission. Buildings design, transportation systems, and open areas significantly affect the sustainability of the urban level. Buildings contribute to the CO2 emissions by 43%, while the transportation share is 32% [10]. Therefore, cities should be designed in a way that minimizes the GHG averages and pollution percentages. The new cities should be designed to keep their inhabitants healthy, secure, and happy. For this aim, a neighborhood must become greener and robust, with a stable ecosystem. Our built environment at the present time suffers enough and an integrated approach is urgently needed. Successful solutions depend on understanding the relationship among the involved sustainability elements; environmental, historical, social, and economic. The solutions should start from the individual building to the block, neighborhood, district, city, region, and up towards the globe. Furthermore, adopting active urban design strategies, such as using renewable energy (PV) solar panels at the urban level, will enhance the sustainability on an urban and city scale [11].

The terms "Neighborhood" or "Community" refer to a number of residential units and the related facilities that serve the resident's needs [12]. Livable, sustainable neighborhoods are one of the determining and essential factors for developing sustainable environments [13]. The sustainable neighborhood is a neighborhood that integrates the three sustainability pillars "Environment, Economy and Society". From the social aspect, providing the required open areas, landscaped areas, playgrounds, and community facilities will encourage sociality and people communications [14]. From the economic aspect, those sustainable neighborhoods that provide all the services and facilities will create livable, healthy independent communities that will have a positive effect on the individuals and the whole society [15]. However, "The Sustainable Urban Design" provides a high level of sustainability and efficiency in terms of major urban design dimensions, such as livability, land use, transportation, buildings design, landscaped areas, and environmental performance [16,17]. However, "The Sustainable Urban Design" provides a high level of sustainability and efficiency in terms of major urban design dimensions, such as liveability, land use, transportations, buildings design, landscaped areas, and environmental performance.

Urban sustainability is significant to the future of humans; it directly affects people's lifestyle, time, effort, health, wellbeing and welfare [18]. Transportation, resources conservation, indoor, and outdoor thermal comfort represent some of the sustainable urban

design factors that have a direct effect on the livable community. The major challenge for the urban designer is to improve and optimize the relation among the three factors in urban geometry; density, movement and recourses. Sustainability at the urban level could be achieved through optimizing the three aspects of, and finding the best design for, the neighborhood, district and city [19]. The urban areas and communities include buildings, open and green spaces, water features and road networks. These are the urban design elements that should be organized in a way that provides vitality and improves the people's lifestyle [20].

Creating a liveable environment is one of the sustainable urban design principles, and the level of urban liveability could be considered important in achieving sustainability in the urban environment. Urban sustainability can be obtained by creating a liveable community, neighbourhood, and city [20]. Urban liveability covers a number of factors; it is a multi-dimensional construct that includes accessibility, number of public parks and open spaces, walkability, transportation planning, urban density, and land use diversity, all of which are design elements that could be improved to achieve high levels of liveability and sustainability [21]. However, it is difficult to define and measure the concept of urban liveability, and set some principles for liveability measurements, such as safety, equity, and continuity [12]. Moreover, when it comes to accessibility and inclusiveness of the previously mentioned indicators [22], accessibility, land use diversity, providing parks and green areas are of the strategies used when planning a sustainable neighborhood. Passive design also has an effective role in achieving a green and sustainable community by offering recourses efficiency [23,24]. The crucial roles of the green space on ecosystem have already been proven by some researchers [25]. The leakage in accessibility to these areas and other community services affects the community sustainability level; ensuring a good accessibility will improve the community sustainability, and this consequently improves the community social life [25]. The service within the green areas, including sport services, has an impressive impact on peoples' wellbeing, and is resulting with good social relations among the residences [25]. It has been proven that the design and the architecture of the buildings should collaborate with surrounding nature to create a harmony between the outdoor and indoor spaces. The concept of human community should be designed to positively influence the human behavior, health and culture [26]. Other than that, land use diversity is another factor that forms a sustainable community. Land use diversity and ensuring a good accessibility to the daily required services would improve peoples' lifestyle from one side, and have a positive effect on resource saving from the other side [27]. The reduction in the use of transportation and vehicle's journey will consequently have a positive impact environmentally, by reducing $CO_2$ emissions [28,29].

Passive design is one of the strategies that the urban planner can adopt for designing a sustainable community according to its direct effect on outdoor and indoor thermal performance [30]. The urban air temperature is rising in all cities around the world, as a result of global warming and the decrease in the natural and greenery area in cities. This rise in outdoor air temperature consequently affects the thermal performance of the inner space environment and increases the indoor air temperature averages [30]. The impact of buildings and urban geometry on the urban heat island phenomena and the outdoor thermal performance has been proven in many studies [31]. Increasing and enhancing the sustainability of our developments is an urgent matter when it comes to facing global warning, resources limitation, and pollution. Implementing the passive and active design elements on buildings and at the urban level represents a part of the solution [32].

Building design, orientation, and block density are of significant effects in development sustainability [33,34]. Creating a desired shading on urban level will have a positive thermal impact on both outdoor and indoor environments. In the hot climate conditions of the UAE, the reduction in air temperature and solar gain due to the orientation can reach 1.8 °C and 13% respectively. [34]. One of the rule of thumb in urban design is the belief that energy consumption decreases when the community or city density increases. This is a challenge to the urban designer to find the best balance between the two variables in

urban planning; density and energy [35]. Furthermore, optimizing the indoor and thermal performance on an urban level will have a positive impact on livability, productivity, and indoor energy consumption [36].

The strong urban structure provides less use or need for transportation and reduces the path, the cycling transportation in the most preferred plan, and different types of transportation plays a significant role in changing the traditional urban structure. The vehicle flow, parking areas, street width, public transportation stations, and many others related to the transit system are the elements that should be well designed to obtain a strong structure [37,38]. Consequently, road planning affects the other urban factors such as gardens and open areas, and playgrounds, which should be counted on during early design stages. The sustainable land use planning is the significant factor in reducing the daily transporting cycle, and increasing walkability as one of the sustainable neighborhood requirements [39]. Furthermore, greenery and landscaped areas could be effective influences on increasing walkability from one side, reducing air temperatures and enhancing outdoor and indoor thermal performances from the other side [40].

The impact of communities and developments has been illustrated previously. The former studies proved the significant impact of sustainable urban design and sustainable developments on reducing the negative environmental effect caused by continuous urbanization. This study aims to contribute to this concept by investigating the potential of improving the performance of one of the Dubai community's performances towards sustainable performance. Hence, a community located in the city of Dubai in the United Arab Emirates (UAE) will be explored, evaluated, and optimized to achieve a sustainable community that follows the sustainable design standards. Dubai is located in the north of the UAE, and extended along the Arabian Gulf with a climate that is known with its humidity during summers, due to its location of the city on Dubai Creek. Generally, the weather in Dubai is sunny most days of the year; in winter, the average temperature is 25 °C, while in summer, the temperature may reach up to 38 °C, with a high percentage of humidity between 20–60%, and a low average of rainy days. The annual air temperature varied between 17 °C in winter and 35 °C by a Dubai weather file generated through the Integrated Environmental Solutions—Virtual Environment (IES-VE) software [41].

## 2. Methods

A case study method has been used to achieve the research aim and objectives. The study focused on exploring the urban sustainability in a selected residential community in Dubai through analysis and evaluation using two separate software packages; (i) IES-VE [41], and (ii) CityCAD [42]. The study will adopt the following steps:

- Analyzing livability of the community through presenting the quantity of land use, services and accessibility.
- Calculating the number of units on the long axis within 15 degrees of the east-west axis.
- Presenting virtual images, plans and reports for the existing case study and the modified scenarios that are suggested in order to optimize the community sustainability.
- Simulating sun path and solar shading analysis using SunCast application.
- Investigating the effect of the suggested modifications on solar gains of the community units in percentage and hours.
- Investigating the effect of the suggested modifications on air temperatures within the community units using ApacheSim application.

In addition to the CityCAD and IES-VE software packages, the community sustainability has been evaluated with the use of the United States Green Building Council (USGBC)'s Leadership in Energy and Environmental Design (LEED) rating system.

In this study, LEED for Neighborhood and Developments (ND), version 4 (2014), has been used [43]. The strategy for the modifications adopted passive urban design solutions to the community master plan, which was also applicable to the existing community. Five of the urban design parameters have been modified according to three scenarios in order to enhance the community livability and thermal performance. The modifica-

tions/scenarios covered; land use, accessibility, and walkability, building design, open and green areas. Moreover, the effect of the modifications has been analyzed to find out the improvements on environmental and thermal performance, as well as through enhancing solar gain performance.

*The Existing Community as a Case Study*

The case study of this research is represented by Al Waha community, which was developed by Dubai Properties Group (DPG). The existing residential community located in "Dubai land" adjacent to the Emirates Road with easy access to Sheikh Mohammed bin Zayed Road through Al Qudra Road. The community is close to the Arabian Ranches, Sport and Motor City communities as key developments (Figure 1). The case study "Al Waha" community consists of 206 semidetached villas where the villas are designed in three types according to bedroom numbers; two, three and four bedrooms (Figure 2). The facilities are very limited in the community, covering swimming pool, playground area, landscape and hardscape.

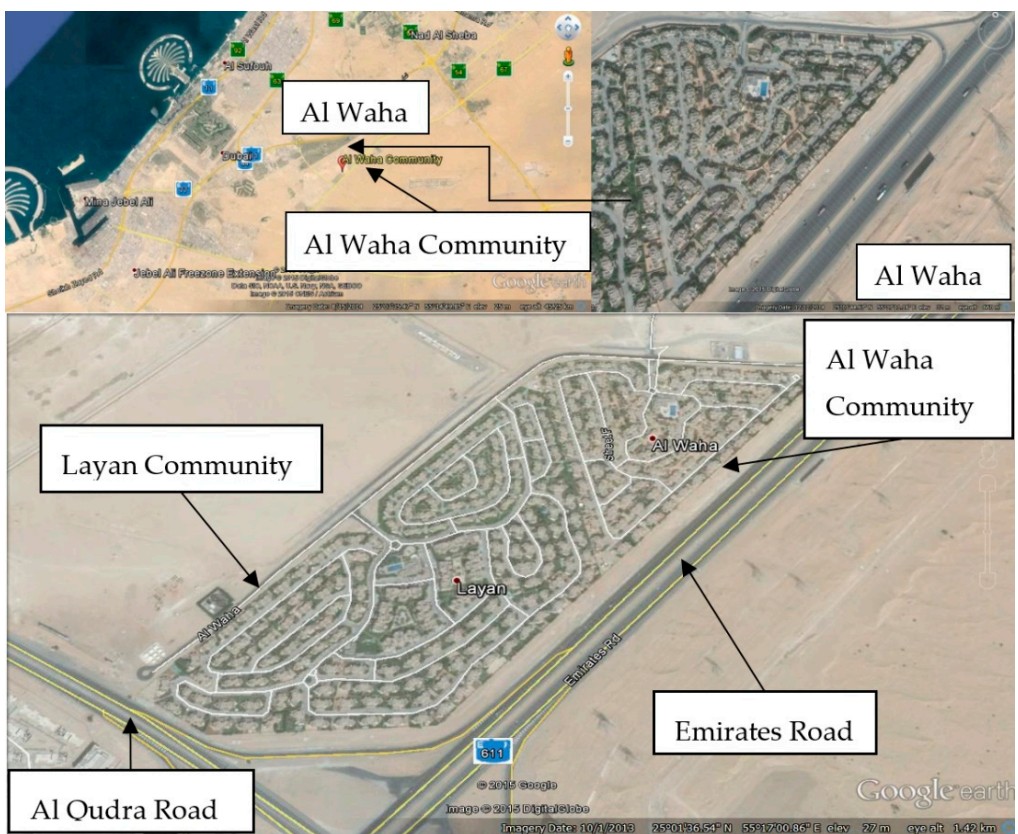

**Figure 1.** The case study location. Al Waha, Dubai [44].

The total area of the community is approximately 130,000 sqm, while the landscape covers 15,200 sqm from the community total area. The neighbor community is the Layan community, from the same developer (DPG), and contains seven G+2 residential buildings and 588 villas, with small facilities such as small shops and a supermarket.

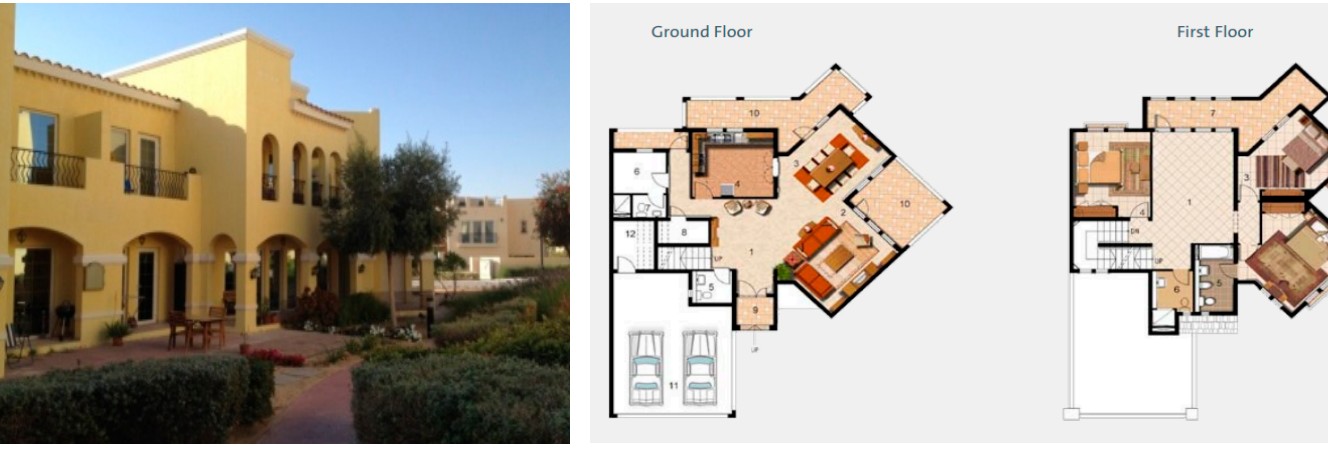

<table>

|  |  |
|---|---|
| (**a**) | (**b**) |

</table>

**Figure 2.** Al Waha community villa view (**a**) and layout (**b**).

## 3. Background for the Analysis

### 3.1. Assessing Livability in the Existing Community

The observation and the assessment during the community site visit, and the use of the CityCAD software for livability analysis showed that there are weaknesses in many livability aspects. The major weakness is in land use diversity, as the existing land use variety is very limited. The community consists of three types of semidetached villas, playground area and a communal swimming pool, with hard and soft landscape. There is a clear absence of many services required, such as supermarket, laundry, pharmacy, school, healthcare center and amenity facilities. The livability analysis of the existing case study using CityCAD shows the average distance from community dwellings to some services and facilities (Table 1).

**Table 1.** Assessing this existing case study services and the adopted scenarios with additional services, namely new services.

| | Average Distance (m) | | |
|---|:---:|:---:|:---:|
| **Services** | **(Existing Case Study)** | **Scenario One** | **Scenario Two** |
| Green Spaces | 30 | 30 | 30 |
| Parking Spaces | 10 | 10 | 10 |
| Playground | 50 | 50 | 50 |
| Public Space and Swimming Pool | 55 | 55 | 55 |
| Shops | 1000 | 200 | 300 |
| Super market | 1000 | 200 | 300 |
| Hot Food and Takeaway | 1000 | 200 | 300 |
| Pharmacy | 8000 | 200 | 300 |
| Educational Services | 8000 | 200 | 300 |
| Metro Station (Emirates) | 15,000 | 15,000 | 15,000 |
| Shopping Mall (Emirates | 15,000 | 15,000 | 15,000 |
| Hospitals | 12,000 | 12,000 | 12,000 |
| | Average Distance (m) | | |
| **New Services** | **(Existing Case Study)** | **Scenario One** | **Scenario Two** |
| Assembly and Leisure | 8000 | 200 | 300 |
| Laundry | 8000 | 200 | 300 |
| Restaurant and Cafe | 8000 | 200 | 300 |
| Financial Services | 8000 | 200 | 300 |

The community, as a gated community, provides a good level of safety as one of the livability requirements [39]. On the other hand, the only one access through the Emirates Road indicated some weakness in accessibility, which could be enhanced and optimized by providing more than one access to improve transition and movement.

### 3.2. Assessing Thermal and Environmental Performance in the Existing Community

Analysis he community layout using integrated environmental solution—virtual environment (IES-VE) software, and adopting a sun path application and unit orientation showed that only 40% of the units are extended along the East-West axis. The benefit of the orientation along the East-West axis is to obtain a minimum amount of solar exposure, as the long facade is facing the North–South axis [43]. The IES-VE software was used to analyze the community shading performance and solar gains through the SunCast application. Figure 3 shows the community layout orientation and the sun path on a summer day, 1 June.

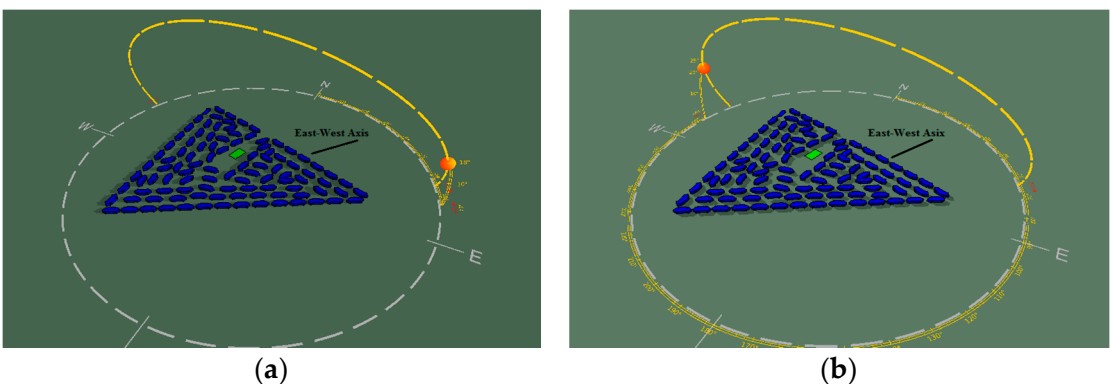

(**a**)　　　　　　　　　　　　　　　　　　　(**b**)

**Figure 3.** The Sun Path analysis during morning and evening along the East-West axis. (**a**) 1 June, 6:00 a.m.; (**b**) 1 June, 6:00 p.m.

The community plot is a triangle shape and one edge of the community plot is extended along the East-West axis, but only 40% of the units extend along the same direction. Thus, the community urban plan would be more sustainable if the units were arranged parallel to the side along the East-West axis in an early planning stage. Yet, it was observed that the compacted form provides more shading and less exposure to solar radiation for the inner units compared to the outer units (Figure 4).

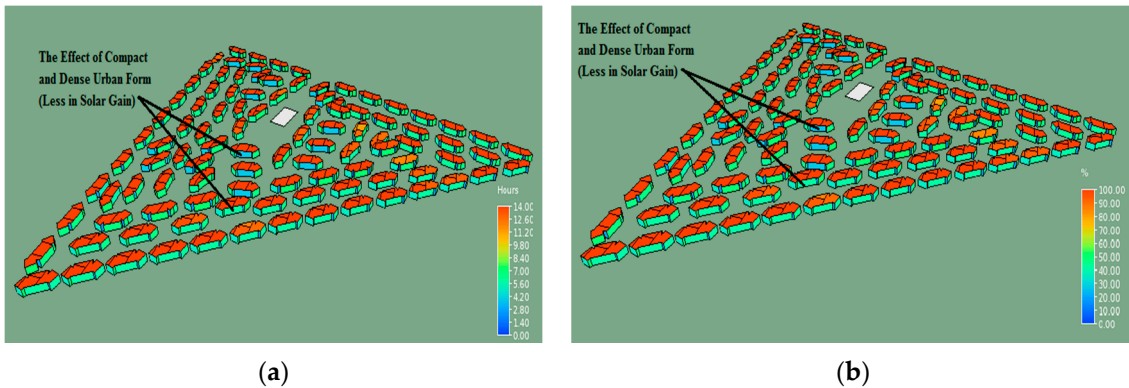

(**a**)　　　　　　　　　　　　　　　　　　　(**b**)

**Figure 4.** Solar attitude in the compact form community during 1 June. (**a**) Solar Gain in Hours; (**b**) Solar Gain in Percentage.

Furthermore, the thermal and environmental performance of the community could be improved through increasing green areas and planting empty/uncultivated areas, which are about 35% of the community landscape area (Figure 5).

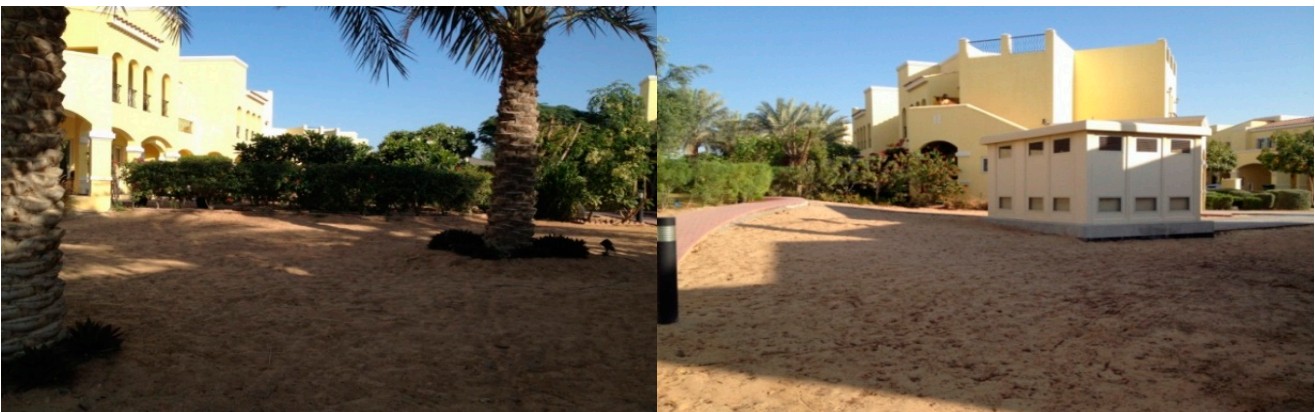

**Figure 5.** Uncultivated areas in the community.

As part of the analysis, exploring the potential of enhancing the community performance towards sustainability, three scenarios were adopted to improve the Al Waha community sustainability. The thinking or the criteria behind these scenarios was to suggest an applicable practice to enhance the community sustainability. Community urban sustainability is improved from two aspects; livability and environmental or thermal performance. Enhancing livability covers a number of parameters; (1) land use, (2) accessibility, (3) walkability, and (4) open and green areas. While the thermal performance parameters represented by improving solar shading and reducing total solar gains through adopting (5) height diversity. The modifications consist of three adopted scenarios to enhance the community sustainability, which are simulated and analyzed by using CityCAD and IES-VE. In addition, using LEED (ND) v4 checklist as an overall and integrated urban sustainability evaluation and assessment tool was to find the sustainability level of the existing and modified case study.

## 4. Results

### 4.1. The Results of the Suggested Scenarios for Enhancing the Community Performance

4.1.1. Scenario One

The community livability could be improved by providing some daily required services such as shops, supermarket, pharmacy, restaurant and cafe, financial services, assembly and leisure, hot food and takeaway. This could be obtained by converting a number of the residential units in the community to provide the missing services. In addition, to add two stories for these units to increase building design diversity (one of LEED's requirement for sustainability) and height diversity as well.

Land use has been improved and a number of facilities were increased by converting some units into services for the daily important and missing facilities, such as adding supermarket, laundry and a pharmacy to be within 300 m–500 m for more than 50% of the community units to fulfill the LEED's land use diversity requirement.

4.1.2. Scenario Two

The second opportunity is enhancing the community services as well as the accessibility, by opening new access to the neighbor community, Layan community, as both of these communities are developed by the same developer (DPG). The new access will allow the residents to benefit from some services that are already existed in Layan community, such as supermarket, bookshop, and small cafe. Furthermore, opening new access to Al Quadra Road will enhance the accessibility and the movement entirely (Figure 6).

Opening new access to Layan community (Scenario Two) would improve the community livability, even though some services are still indicating a weak performance such as educational and medical services. This could be resolved by providing these services

(primary school or medical center) in Layan community, as it is larger in area and has a number (7) of mid-rise buildings, which could be useful for this type of services (Table 1).

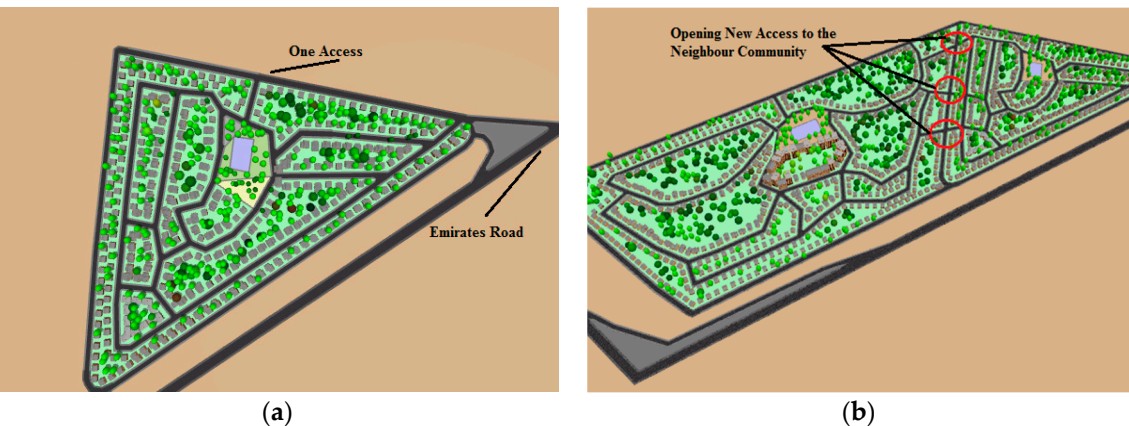

(**a**)  (**b**)

**Figure 6.** (**a**) The existing gated community—one access; (**b**) New access to the neighbor community enhancing access.

Both scenarios 1 and 2 indicated improvement toward a more sustainable setting, as most of the mentioned services are within the LEED ND requirements (i.e., 200–300 m) (Table 1), even though the other services are still at a distance of 12–15 km from the two communities and therefore not able to fulfill the LEED ND sustainability credit requirements.

### 4.1.3. Scenario Three

Enhancing livability could also be achieved by increasing the open spaces and green areas; the open spaces are limited in the community but could be increased when adopting the new access to the Layan community. On the other hand, the green area could be increased by planting the uncultivated areas, which are calculated using site surveys and Google Earth (Pro) [44], while represented in CityCAD by 35% of the landscape area (Figure 7).

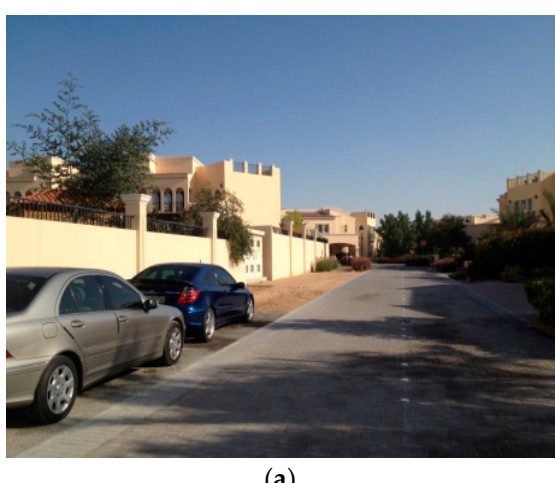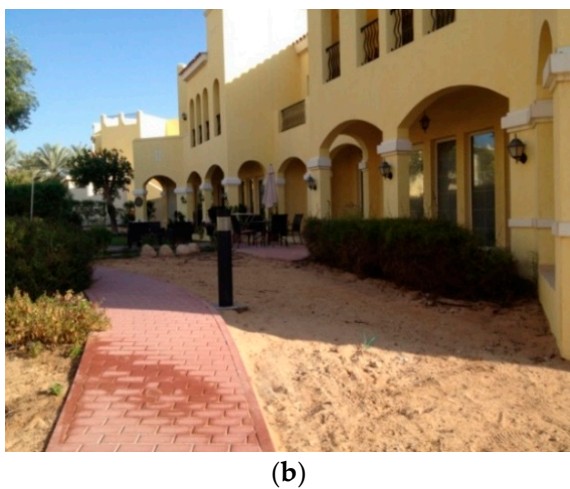

(**a**)  (**b**)

**Figure 7.** (**a**) The existing community walkway beside a road, (**b**) The pedestrian walkway that could be enhanced by adding rubber track, shading devices, and sport equipment.

In addition to increasing the number of trees and adding a green belt alongside the community boundary wall, adding some sports and kids playing equipment to provide the residents and the kids with a place for relaxation and amenity could allow for social communication, while also improving social sustainability (Figure 8).

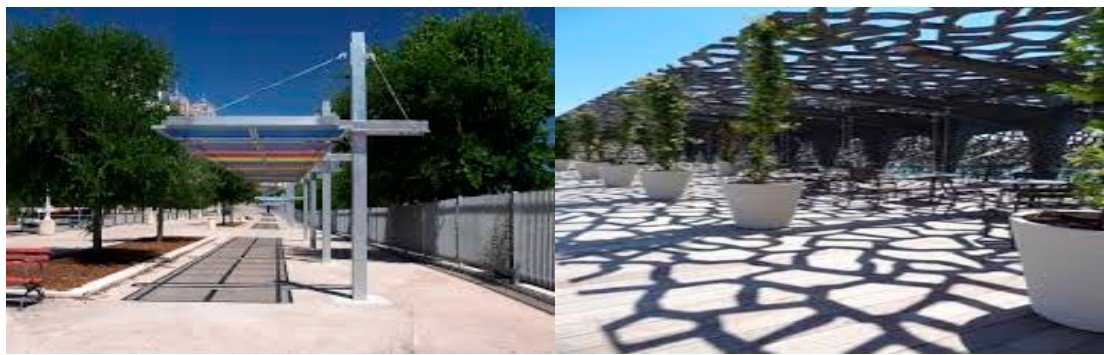

**Figure 8.** Options of shading devices for walkways [45].

Furthermore, walkability could be increased by providing a rubber pathway, shaded walkway [45,46], and a number of benches to encourage people, especially elderly people, to walk and use the community green areas, as encouraging elderly people to walk is one of the social sustainability targets [47,48] (Figure 9).

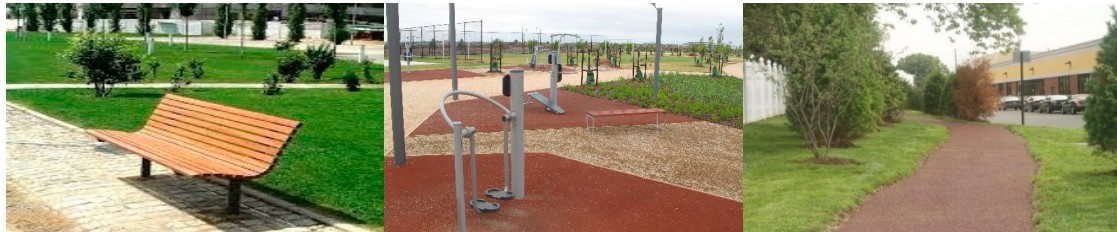

**Figure 9.** Adding benches and enhancing empty areas by using hard and soft landscaping [45,46], and improving walkability for pedestrians by adding rubber walkways (1.1 m width) and outdoor equipment according to the LEED ND requirements

Moreover, adding a green belt around the community could provide more protection and shade areas further to increasing the total number of trees, which has an important role in enhancing the community environmental performance (Figure 10).

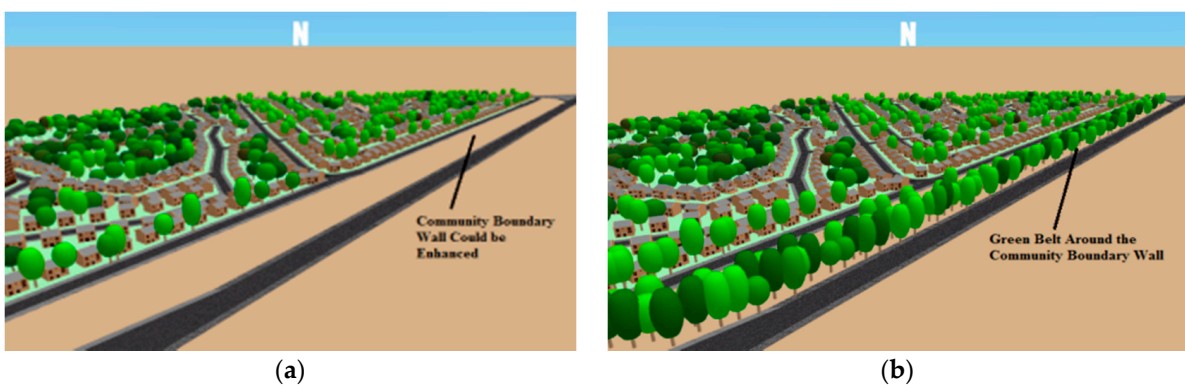

| (a) | (b) |
|:---:|:---:|

**Figure 10.** Planting the community boundary wall [42]. (**a**) Current Setting; (**b**) After Improvements.

### 4.2. The Effects of the Adopted Scenarios on Solar Shading and Solar Gains

Running IES-VE simulations with different heights of community units showed the importance of the height diversity in creating preferable shaded areas for walking people, in addition to the effect of reducing solar gains from the surrounding units. Figure 11 shows that the solar gains of the surrounding units decreased from 100% to 80% (80 h) when applying height diversity.

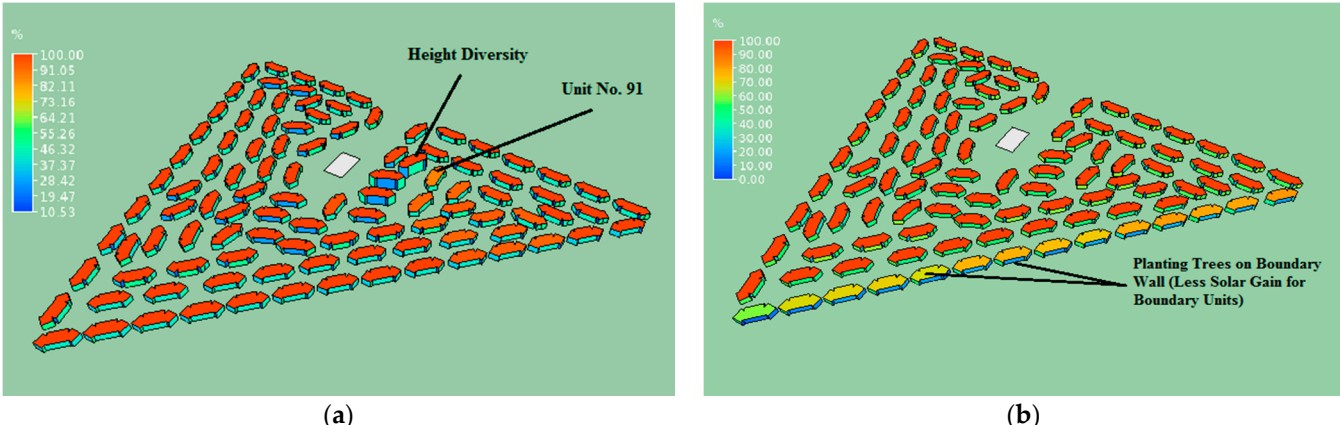

**Figure 11.** (**a**) Solar gains and the effect of diversity in building heights on shading parameters; (**b**) Solar gains and the effect of planting the boundary wall.

The effect of diversity in building heights for creating comfortable outdoor environments was proven by Edward (2010). The researcher explored the benefits of the diversity, dense, and compact form on the outdoor environment, presenting the "Environmental Diversity Map" to show the effect of diversity on the three microclimate parameters; temperature, shading and wind on the outdoor environment [23].

A model using IES-VE SunCast analysis to study the effect of planting height and dense trees along the boundary wall were simulated, as shown in Figure 11b. It is clear that the boundary units adjacent to the boundary wall are varied in solar gains, and the exposure percentage between 50–70% depending on the height and the dense of the trees, with 100% solar gains for the other units.

The effect of the first scenario and the modification in building height diversity analyzed and explored using one of the community units, Unit No. 91. Unit 91 was selected for this analysis due to being oriented toward west direction and has maximum solar gains with 100% in the existing case, and is close to the chosen building, to be converted to serve for the missing services, which is modified and increased in height by adding two more stories (to Unit No. 90), and the adjacent Unit 91 could be therefore less in solar exposure by 20% as it has only 80 h of exposure to the sun (Figure 12).

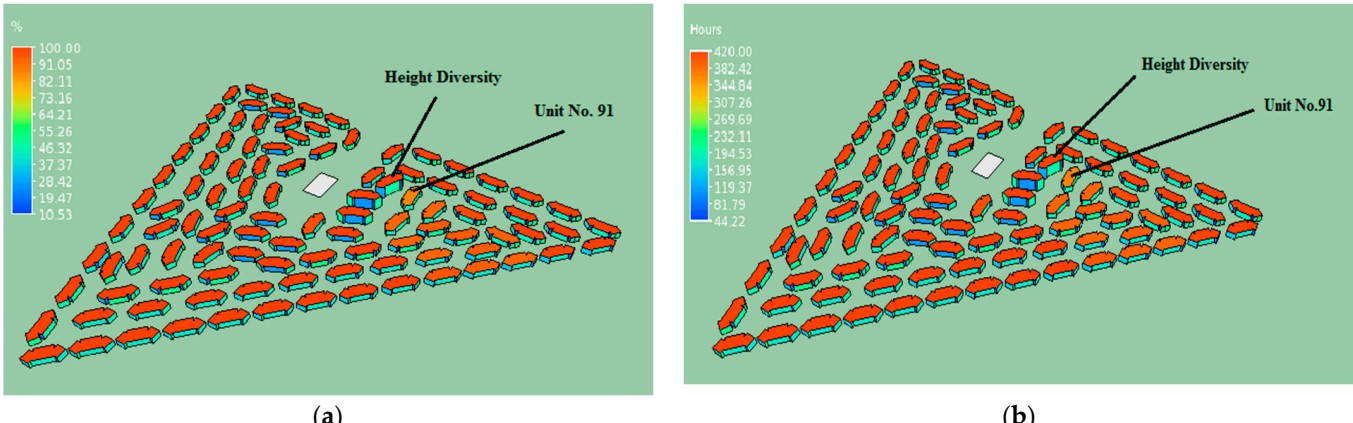

**Figure 12.** Building height diversity and shaded roofs adjacent to the modified building, Unit 91. Less in solar exposure by 20% and 80 h during the month of June. (**a**) SunCast solar shading analysis in percentage; (**b**) SunCast solar shading analysis in hours.

Moreover, using ApacheSim application within the IES-VE software showed that there is a reduction in solar gains by 20% for the modified case compared to the existing case

(Unit 91), and the solar exposure and solar gain hours are 80 h less in total during the month of June, with a reduction from 420 h to 340 h. Furthermore, the reduction in solar gains for Unit 91 showed a comparison between the existing case study (Unit 91) and the new shaded same unit with a reduction of 18.5% in total solar gains on 1 June.

### 4.3. The Community Assessment Using LEED (ND) Rating Tool

LEED Neighborhood as a sustainability assessment tool was used to evaluate the community sustainability or greenness, where the tool rating system consists of five categories, and each category covers a number of requirements. The requirements divided into mandatory requirements and optional requirements; for optional requirements, LEED allocates a number of points or credits for each category, as shown in Table 2.

**Table 2.** LEED (ND) allocated points.

| Requirements | Points |
| --- | --- |
| Smart Location and Linkage | 28 Points |
| Neighborhood Pattern and Design | 41 Points |
| Green Infrastructure and Buildings | 31 Points |
| Innovation and Design Process | 6 Points |
| Regional Priority Credits | 4 Points |

The total number of points collected indicates the level of each community sustainability according to the following scale: Certified 40–49, Silver 50–59, Gold 60–69, Platinum 80+.

Using the LEED Neighborhood and Developments checklist to assess Al Waha community through each of the five categories resulted in the following:

- Smart Location and Linkage: The community fulfill all the five required items and obtain only 11 out of 25 points allocated to this category, as there is a clear weakness in community linkage and accessibility.
- Neighborhood Pattern and Design: With regards to the community design, the three required items related to the pattern are available in community design, and the community collected 16 out of 41 credit points. The demerits of the community design indicated in land use diversity, building design and affordability, and open and assembly areas.
- Green Infrastructure and Buildings: This category indicates a weakness in following the sustainable design requirements related to water and energy efficiency, solar orientation, the use of renewable energy, and the requirement of green building certification; only 4 points obtained out of 31 total points allocated.
- Innovation and Design Process: This category provides points to the new sustainable innovation not addressed in LEED, and none of the six innovation points were able to be collected.
- Regional Priority Credits: This category related to the regional practices and material, and only one regional point out of four was collected.

In total, the community collected only 32 points. This result indicates a low level of sustainability, and the community could not be certified as a green or sustainable community according to the LEED (ND) assessment tool. Generally, the weaknesses are indicated in land use, facilities, accessibility, transportation system, and water and energy efficiency.

The modified case study according to the three scenarios was assessed using the LEED assessment tool, and the results for each category are as follows (Figure 13):

- Smart Location and Linkage: In addition to the 11 points that were collected from the existing case study assessment, providing new access to the community to the main road and new access to the Layan community, adding four additional points, to be 15 points in total for this category.

- Neighborhood Pattern and Design: This category has been improved to collect 24 points as the modified community offer a required diversity in land use and building affordability, further to enhancing walkability, green and open areas.
- Green Infrastructure and Buildings: This category indicates a weakness in following the sustainable design requirement, only four points obtained out of 31 total points allocated.
- Innovation and Design Process: This category provides points to the new sustainable innovation not addressed in LEED, and none of the six innovation points were able to be collected.
- Regional Priority Credits: This category was related to the regional practices and material, and only one regional point out of four was collected.

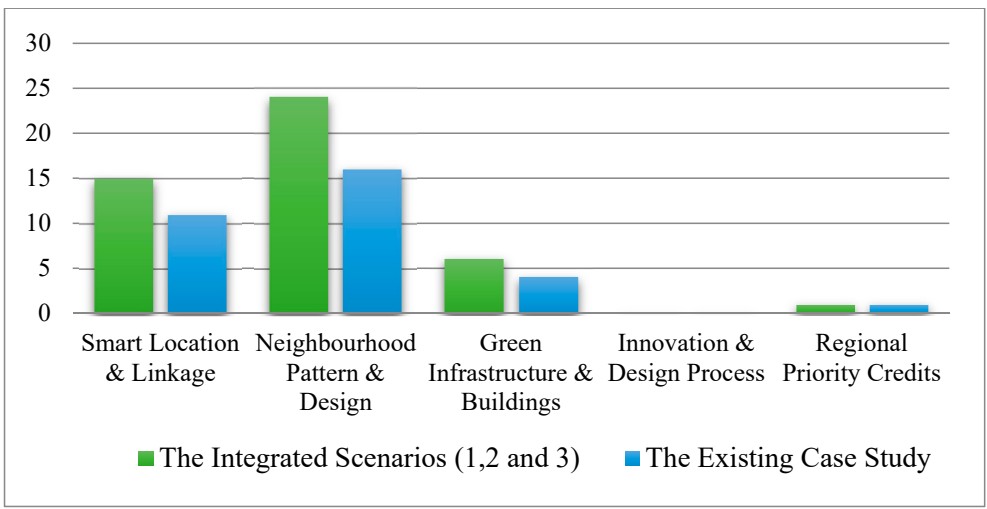

**Figure 13.** LEED assessment, comparison between existing and modified case study.

Moreover, it is worth mentioning that the aspect of building density has helped to improve and optimize relations among the factors in urban geometry [4,6], and therefore emphasizing on this aspect as a key solution in this analyzed context, which should also be taken into account in different climatic zones.

In total, the community collected 46 points and could be certified as a green community (Appendix A). This result shows the effect of the applicable practices and modifications to enhance the community sustainability.

## 5. Discussion

This study aimed to assess and optimize sustainability on the urban scale by selecting and evaluating one of Dubai's communities. The study investigated the potential of enhancing the case study neighborhood towards sustainability according to the sustainability rating system standards. The selected case study community was evaluated, and the likely weakness was indicated in the livability aspects covering land-use diversity, accessibility, transportation system, green and landscape area, and energy efficiency. Moreover, assessing the community showed a capability of enhancing the thermal performance, in addition to the community livability.

Different scenarios were applied in order to improve the community's sustainability on two levels; livability and thermal performance (Figure 13). The adopted scenarios targeted improving the livability by enhancing (1) walkability, (2) accessibility, (3) facilities, and (4) land use diversity [5,13,17]. The thermal performance has been enhanced by increasing the (1) shading effect (2) and improving greenery and landscape areas [34,39]. However, applying the suggested scenarios resulted in improving the community sustainability and resulted in upgrading the community rating level by increasing the number of points that can be achieved by applying LEED (ND) standards [43]. The community gained an additional 14 points,

and resulted to be certified as a sustainable community. The study proved the capability of enhancing the sustainability level of the communities by implementing sustainable design strategies at the urban level. Improving the community performance will result in a positive impact on the total environmental performance [34,39,43].

## 6. Conclusions

Recently, and as part of the future vision, there is a strong trend toward passive design as an effective part in sustainable design. This is a result of real consciousness in limited resources, global warning and pollution problems, where sustainability is the only solution for sustaining our very future in the world. In line with this context, this study aimed to explore and improve a neighborhood in Dubai, UAE toward a more livable, sustainable community.

The Al Waha community in Dubai was selected for the analysis, and three scenarios were adopted for developing and obtaining more sustainable community. Analysis of the community covered two of the sustainable urban design dimensions (1) livability and (2) thermal and environmental performance. Livability analysis of the existing case study showed some clear weaknesses in land use diversity, accessibility, walkability, landscaped area and building design diversity. These weaknesses were covered in the adopted scenarios, through analyses using the two software packages, CityCAD and IES-VE, and the LEED (ND) checklist, showing clear improvements in all mentioned parameters. Community improvement strategy and adopted scenarios covered a number of urban design parameters including; (1) land use diversity, (2) accessibility, (3) walkability, (4) open public area and green spaces, and (5) building height and design variety, which directly affected environmental or thermal performance parameters covering solar gains and air temperatures. The effects of the adopted scenarios (with modified computer models) on solar gains and thermal performances have been studied and analyzed using the IES-VE applications SunCast and ApacheSim—Vista Pro. The results showed that the livability level of the community was increased by enhancing the land use diversity, accessibility, walkability, building height diversity and the green areas. The aspect of building density has played a key role in the analysed context to help the community livability. Furthermore, the effect of the adopted scenarios to enhance the community livability showed a clear and positive effect on environmental and thermal performance by increasing the shading effect and reducing indoor solar gains and air temperatures. Finally, the modified community that integrated in the three adopted scenarios have been evaluated using the LEED Neighborhood and Developments (ND) assessment tool v4, and the community was able to be certified as a "Sustainable Green Community" through implementing all of the applicable practices.

**Author Contributions:** S.S. and H.A. conceived and designed the concept and the paper outline; S.S. conducted the analyses and wrote the paper; and H.A. supervised, provided direction, sources, comments, and major edits to the paper. All authors have read and agreed to the published version of the manuscript.

**Funding:** This research received no external funding.

**Institutional Review Board Statement:** Not applicable.

**Informed Consent Statement:** Not applicable.

**Data Availability Statement:** Not applicable.

**Conflicts of Interest:** The authors declare no conflict of interest.

# Appendix A. LEED Neighborhood & Development, (ND) V4 Checklist

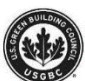

**LEED v4 for Neighbourhood Development Plan**

**Project Name: "Al Waha Community", Dubai, UAE**
**The Existing Case Study**

| Yes | ? | No | | | |
|---|---|---|---|---|---|
| **11** | **0** | **0** | **Smart Location & Linkage** | | **28** |
| Y | | | Prereq | Smart Location | Required |
| Y | | | Prereq | Imperiled Species and Ecological Communities | Required |
| Y | | | Prereq | Wetland and Water Body Conservation | Required |
| Y | | | Prereq | Agricultural Land Conservation | Required |
| Y | | | Prereq | Floodplain Avoidance | Required |
| 5 | | | Credit | Preferred Locations | 10 |
| | | | Credit | Brownfield Remediation | 2 |
| 3 | | | Credit | Access to Quality Transit | 7 |
| 1 | | | Credit | Bicycle Facilities | 2 |
| | | | Credit | Housing and Jobs Proximity | 3 |
| 1 | | | Credit | Steep Slope Protection | 1 |
| | | | Credit | Site Design for Habitat or Wetland and Water Body Conservation | 1 |
| | | | Credit | Restoration of Habitat or Wetlands and Water Bodies | 1 |
| 1 | | | Credit | Long-Term Conservation Management of Habitat or Wetlands and Water Bodies | 1 |
| **16** | **0** | **0** | **Neighbourhood Pattern & Design** | | **41** |
| Y | | | Prereq | Walkable Streets | Required |
| Y | | | Prereq | Compact Development | Required |
| Y | | | Prereq | Connected and Open Community | Required |
| 4 | | | Credit | Walkable Streets | 9 |
| 4 | | | Credit | Compact Development | 6 |
| | | | Credit | Mixed-Use Neighbourhoods | 4 |
| 3 | | | Credit | Housing Types and Affordability | 7 |
| 1 | | | Credit | Reduced Parking Footprint | 1 |
| 1 | | | Credit | Connected and Open Community | 2 |
| | | | Credit | Transit Facilities | 1 |
| | | | Credit | Transportation Demand Management | 2 |
| 1 | | | Credit | Access to Civic & Public Space | 1 |
| | | | Credit | Access to Recreation Facilities | 1 |
| 1 | | | Credit | Visitability and Universal Design | 1 |
| | | | Credit | Community Outreach and Involvement | 2 |
| | | | Credit | Local Food Production | 1 |
| 1 | | | Credit | Tree-Lined and Shaded Streetscapes | 2 |
| | | | Credit | Neighbourhood Schools | 1 |
| **4** | **0** | **0** | **Green Infrastructure & Buildings** | | **31** |
| Y | | | Prereq | Certified Green Building | Required |
| Y | | | Prereq | Minimum Building Energy Performance | Required |
| Y | | | Prereq | Indoor Water Use Reduction | Required |
| Y | | | Prereq | Construction Activity Pollution Prevention | Required |
| | | | Credit | Certified Green Buildings | 5 |
| | | | Credit | Optimize Building Energy Performance | 2 |
| 1 | | | Credit | Indoor Water Use Reduction | 1 |
| 1 | | | Credit | Outdoor Water Use Reduction | 2 |
| | | | Credit | Building Reuse | 1 |
| | | | Credit | Historic Resource Preservation and Adaptive Reuse | 2 |
| 1 | | | Credit | Minimized Site Disturbance | 1 |
| | | | Credit | Rainwater Management | 4 |
| | | | Credit | Heat Island Reduction | 1 |
| | | | Credit | Solar Orientation | 1 |
| | | | Credit | Renewable Energy Production | 3 |
| | | | Credit | District Heating and Cooling | 2 |
| | | | Credit | Infrastructure Energy Efficiency | 1 |
| 1 | | | Credit | Wastewater Management | 2 |
| | | | Credit | Recycled and Reused Infrastructure | 1 |
| | | | Credit | Solid Waste Management | 1 |
| | | | Credit | Light Pollution Reduction | 1 |
| **0** | **0** | **0** | **Innovation & Design Process** | | **6** |
| | | | Credit | Innovation | 5 |
| | | | Credit | LEED® Accredited Professional | 1 |
| **1** | **0** | **0** | **Regional Priority Credits** | | **4** |
| 1 | | | Credit | Regional Priority Credit: Region Defined | 1 |
| | | | Credit | Regional Priority Credit: Region Defined | 1 |
| | | | Credit | Regional Priority Credit: Region Defined | 1 |
| | | | Credit | Regional Priority Credit: Region Defined | 1 |
| **32** | **0** | **0** | **PROJECT TOTALS (Certification estimates)** | | **110** |

**Certified:** 40-49 points, **Silver:** 50-59 points, **Gold:** 60-79 points, **Platinum:** 80+ points

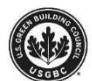

**LEED v4 for Neighbourhood Development Plan**

**Project Name: "Al Waha Community", Dubai, UAE**
**The Integrated Scenarios and Modified Case**

| Yes | ? | No | | | |
|---|---|---|---|---|---|
| **15** | **0** | **0** | **Smart Location & Linkage** | | **28** |
| Y | | | Prereq | Smart Location | Required |
| Y | | | Prereq | Imperiled Species and Ecological Communities | Required |
| Y | | | Prereq | Wetland and Water Body Conservation | Required |
| Y | | | Prereq | Agricultural Land Conservation | Required |
| Y | | | Prereq | Floodplain Avoidance | Required |
| 6 | | | Credit | Preferred Locations | 10 |
| | | | Credit | Brownfield Remediation | 2 |
| 5 | | | Credit | Access to Quality Transit | 7 |
| 2 | | | Credit | Bicycle Facilities | 2 |
| | | | Credit | Housing and Jobs Proximity | 3 |
| 1 | | | Credit | Steep Slope Protection | 1 |
| | | | Credit | Site Design for Habitat or Wetland and Water Body Conservation | 1 |
| | | | Credit | Restoration of Habitat or Wetlands and Water Bodies | 1 |
| 1 | | | Credit | Long-Term Conservation Management of Habitat or Wetlands and Water Bodies | 1 |
| **24** | **0** | **0** | **Neighbourhood Pattern & Design** | | **41** |
| Y | | | Prereq | Walkable Streets | Required |
| Y | | | Prereq | Compact Development | Required |
| Y | | | Prereq | Connected and Open Community | Required |
| 4 | | | Credit | Walkable Streets | 9 |
| 4 | | | Credit | Compact Development | 6 |
| 2 | | | Credit | Mixed-Use Neighbourhoods | 4 |
| 5 | | | Credit | Housing Types and Affordability | 7 |
| 1 | | | Credit | Reduced Parking Footprint | 1 |
| 2 | | | Credit | Connected and Open Community | 2 |
| | | | Credit | Transit Facilities | 1 |
| | | | Credit | Transportation Demand Management | 2 |
| 1 | | | Credit | Access to Civic & Public Space | 1 |
| 1 | | | Credit | Access to Recreation Facilities | 1 |
| 1 | | | Credit | Visitability and Universal Design | 1 |
| 1 | | | Credit | Community Outreach and Involvement | 2 |
| | | | Credit | Local Food Production | 1 |
| 1 | | | Credit | Tree-Lined and Shaded Streetscapes | 2 |
| 1 | | | Credit | Neighbourhood Schools | 1 |
| **6** | **0** | **0** | **Green Infrastructure & Buildings** | | **31** |
| Y | | | Prereq | Certified Green Building | Required |
| Y | | | Prereq | Minimum Building Energy Performance | Required |
| Y | | | Prereq | Indoor Water Use Reduction | Required |
| Y | | | Prereq | Construction Activity Pollution Prevention | Required |
| | | | Credit | Certified Green Buildings | 5 |
| 2 | | | Credit | Optimize Building Energy Performance | 2 |
| 1 | | | Credit | Indoor Water Use Reduction | 1 |
| 1 | | | Credit | Outdoor Water Use Reduction | 2 |
| | | | Credit | Building Reuse | 1 |
| | | | Credit | Historic Resource Preservation and Adaptive Reuse | 2 |
| 1 | | | Credit | Minimized Site Disturbance | 1 |
| | | | Credit | Rainwater Management | 4 |
| | | | Credit | Heat Island Reduction | 1 |
| | | | Credit | Solar Orientation | 1 |
| | | | Credit | Renewable Energy Production | 3 |
| | | | Credit | District Heating and Cooling | 2 |
| | | | Credit | Infrastructure Energy Efficiency | 1 |
| 1 | | | Credit | Wastewater Management | 2 |
| | | | Credit | Recycled and Reused Infrastructure | 1 |
| | | | Credit | Solid Waste Management | 1 |
| | | | Credit | Light Pollution Reduction | 1 |
| **0** | **0** | **0** | **Innovation & Design Process** | | **6** |
| | | | Credit | Innovation | 5 |
| | | | Credit | LEED® Accredited Professional | 1 |
| **1** | **0** | **0** | **Regional Priority Credits** | | **4** |
| 1 | | | Credit | Regional Priority Credit: Region Defined | 1 |
| | | | Credit | Regional Priority Credit: Region Defined | 1 |
| | | | Credit | Regional Priority Credit: Region Defined | 1 |
| | | | Credit | Regional Priority Credit: Region Defined | 1 |
| **46** | **0** | **0** | **PROJECT TOTALS (Certification estimates)** | | **110** |

**Certified:** 40–49 points, **Silver:** 50–59 points, **Gold:** 60–79 points, **Platinum:** 80+ points

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
