# Peer review of "Sustainability at an Urban Level: A Case Study of a Neighborhood in Dubai, UAE"

_sustainability, doi:10.3390/su13084355_

Round 1

Reviewer 1 Report

This study investigated the sustainability of the region by investigating the livability and environmental performance. The evaluation is carried out based on the LEEDs.

  1. The objectives are not organized properly (183~192): some are rather research step or task than the objectives 
  2. reference is not organized properly in text: e.g., numbering
  3. The results can be shown with some figure on top of the table to improve the readability of the work.
  4. The results can be organized with bullet point to improve the clarity of the study.

livability by enhancing walkability, accessibility, facilities, and land use 443 diversity

environmental performance has been enhanced by increasing the 444 shading effect and improving greenery and landscape areas

Author Response

This study investigated the sustainability of the region by investigating the livability and environmental performance. The evaluation is carried out based on the LEEDs.

  • The objectives are not organized properly (183~192): some are rather research step or task than the objectives.

The text has been revised according to the reviewer's comment to avoid any confusion.

  • Reference is not organized properly in text: e.g., numbering.

The references have been reorganized properly.

  • The results can be shown with some figure on top of the table to improve the readability of the work.

Some figures with unreadable text have been removed and replaced with the tables, new tables added.

  • The results can be organized with bullet point to improve the clarity of the study.

“Livability by enhancing walkability, accessibility, facilities, and land use 443 diversity

environmental performance has been enhanced by increasing the 444 shading effect and improving greenery and landscape areas.”

The results have been organised with bullet points as recommended by the reviewer.

Reviewer 2 Report

The article, "Sustainability at an Urban Level: A Case Study of a Neighborhood in Dubai, UAE ", is intended to examine two major factors in neighborhood sustainable design, such as livability and thermal performance.

The research topic is relevant and interest in this area of sustainability at the city level is growing, as has been noted in the introduction, especially with regard to expected climate change. The background for the study, including a motivation supported by a relevant literature review, is well presented initially. However, the selected case study should be presented in more detail, and some significant weaknesses should be excluded before publication can be considered:

  • Keywords: repetitions, such as Sustainability/Neighborhood Sustainability, where the latter is more appropriate, should be avoided.
  • Other repetitions, such as in the introduction Environment/Economy/Society, should be avoided and at the same time the most important studies that support the research question should be emphasized.
  • The aim of the study is to examine livability and thermal performance, as mentioned in the abstract and the introduction, while thermal performance is inadequately investigated. Instead, environmental performance is discussed later, for instance in discussion. It is a difference.
  • Almost all presented figures’ quality (except photos) is not acceptable. The quality of graphics is too low, and it is impossible to read the information presented. In addition, the information is not presented in sufficient detail, does not correspond well with the descriptions (texts) that follow, and it is therefore impossible to evaluate its relevance.
  • Some information presented in Chapter The Existing Community as a Case Study belongs to discussion or conclusions and should be moved there.
  • In general: the results of calculations/simulations are inadequately presented. Without some relevant detailed data presenting specific results, it is impossible to evaluate accuracy, suitability and relevance of the study. There are only final results, which belong rather to discussion and conclusions, presented in the article in Chapter Analysis and Results.
  • The authors must be more aware of the way information/data is presented, for example units are missing in Table 1.
  • As a consequence of inadequately/inconsistently presented results of the analysis, it is impossible to evaluate the discussion and the relevance of the conclusions.

Respecting the efforts of the authors and in their interests, I firmly believe that the article should be significantly improved and reviewed again before publication, especially as it touches upon an important topic.

Author Response

  • Keywords: repetitions, such as Sustainability/Neighborhood Sustainability, where the latter is more appropriate, should be avoided.

The Keywords have been revised according to the review comment.

  • Other repetitions, such as in the introduction Environment/Economy/Society, should be avoided and at the same time the most important studies that support the research question should be emphasized.

The repetitions have been removed.

  • The aim of the study is to examine livability and thermal performance, as mentioned in the abstract and the introduction, while thermal performance is inadequately investigated. Instead, environmental performance is discussed later, for instance in discussion. It is a difference.

This is mainly related to the overlapping between the impact of the suggested scenarios on both livability and thermal and environmental performance, this is the reason for using the "Environmental" term. However, this point has been clarified according to the reviewer's comment to avoid any confusion.

  • Almost all presented figures’ quality (except photos) is not acceptable. The quality of graphics is too low, and it is impossible to read the information presented. In addition, the information is not presented in sufficient detail, does not correspond well with the descriptions (texts) that follow, and it is therefore impossible to evaluate its relevance.

The illustrated figures were extracted from the CityCad urban simulation software. The presented figures have the maximum resolution that can be achieved. Therefore, the author removed the unreadable figures and the same information has been presented in the tables.

  • Some information presented in Chapter “The Existing Community as a Case Study” belongs to discussion or conclusions and should be moved there.

The information has been moved to the analysis and discussion sections.

  • In general: the results of calculations/simulations are inadequately presented. Without some relevant detailed data presenting specific results, it is impossible to evaluate accuracy, suitability and relevance of the study. There are only final results, which belong rather to discussion and conclusions, presented in the article in Chapter.

The results provided by the simulation process have been presented in the analysis and results section and results in detail, more data and details were added complying with the reviewer comment. A separate discussion section addressed all the results and the integrated impact of the suggested scenarios on the community performance.

  • The authors must be more aware of the way information/data is presented, for example units are missing in Table 1.

The data presented has been rechecked and the units have been added to the table.

  • As a consequence of inadequately/inconsistently presented results of the analysis, it is impossible to evaluate the discussion and the relevance of the conclusions.

The results have been represented consistently and more details have been added complying with the reviewer’s comment.

Respecting the efforts of the authors and in their interests, I firmly believe that the article should be significantly improved and reviewed again before publication, especially as it touches upon an important topic.

We appreciate the reviewer’s constructive comments, which helped us further improve our article to a better standard.

Round 2

Reviewer 2 Report

The article, "Sustainability at an Urban Level: A Case Study of a Neighborhood in Dubai, UAE " is intended to examine two major factors in neighborhood sustainable design, such as livability and thermal performance.

The authors have made significant improvements and this article should be published as this research topic is important and will be discussed more frequently at a global level in the future. However, the authors may consider some minor improvements:

  • Keywords: Neighborhood Sustainability, Livability … or Sustainability & Livability of Neighborhoods; then Solar shading or Energy efficient/Sustainable solar shading; then UAE should be explained, also as a keyword before the abbreviation.
  • Abstract: it would be beneficial to add specific/most important assumptions/founds from the study, relating to the results/conclusion, rather than just the conclusion that the study showed that the recommended actions and modifications
  • Materials and Methods or Methods and Software (the methods are the most important; it may only suffice with Methods even if software is mentioned; the use of materials can be misunderstood).
  • Figures: please, pay attention to the arrangement of information accompanying the drawings with the regard to the editing process. Some data may still be difficult to read. If better-quality illustrations cannot be obtained, another way to present the important data may help.
  • Analysis and Results: please, consider to chapters separately Background for the analysis and Results
  • It would be beneficial to emphasize the aspect of building density, particularly in discussion and conclusion, and present it more clearly as it is a key aspect in this analyzed context which should be taken into account in different climatic zones.

I firmly believe that some minor improvements would make the article even more attractive and should be considered by the authors. Then, the article should be published.

Author Response

The article, "Sustainability at an Urban Level: A Case Study of a Neighborhood in Dubai, UAE" is intended to examine two major factors in neighborhood sustainable design, such as livability and thermal performance.

The authors would like to thank the reviewer for their encouragement and constructive feedback.

The authors have made significant improvements and this article should be published as this research topic is important and will be discussed more frequently at a global level in the future. However, the authors may consider some minor improvements:

  • Keywords: Neighborhood Sustainability, Livability … or Sustainability & Livability of Neighborhoods; then Solar shading or Energy efficient/Sustainable solar shading; then UAE should be explained, also as a keyword before the abbreviation.

Keywords have been updated accordingly with the suggestions.

  • Abstract: it would be beneficial to add specific/most important assumptions/founds from the study, relating to the results/conclusion, rather than just the conclusion that the study showed that the recommended actions and modifications

Abstract has been revised to reflect more on the results as suggested.

  • Materials and Methods or Methods and Software (the methods are the most important; it may only suffice with Methods even if software is mentioned; the use of materials can be misunderstood).

Section heading has been updated.

  • Figures: please, pay attention to the arrangement of information accompanying the drawings with the regard to the editing process. Some data may still be difficult to read. If better-quality illustrations cannot be obtained, another way to present the important data may help.

Figures have been rechecked and updated.

  • Analysis and Results: please, consider to chapters separately Background for the analysis and Results

Section headings have been separated and updated accordingly with the suggestion.

  • It would be beneficial to emphasize the aspect of building density, particularly in discussion and conclusion, and present it more clearly as it is a key aspect in this analyzed context which should be taken into account in different climatic zones.

Building density aspect has been emphasised and presented more clearly in both Discussion and Conclusion sections.

I firmly believe that some minor improvements would make the article even more attractive and should be considered by the authors. Then, the article should be published.

The authors appreciate the support of the reviewers.
